# Effect of emancipative values on life satisfaction across different levels of democracy: A cross-national analysis of the World Values Survey

**Guillaume Barbalat** [ORCID]*, **Nicolas Franck**

Centre ressource de réhabilitation psychosociale et de remédiation cognitive, Hôpital Le Vinatier, Pôle Centre rive gauche, UMR, CNRS & Université Claude Bernard Lyon 1, France

* guillaume.barbalat@ch-le-vinatier.fr

## Abstract

Emancipative values – advocating for personal freedoms, equality, and autonomy – are theorized to enhance life satisfaction by fostering empowerment and opportunities. The current study tested whether their impact is context-dependent, influenced by societal norms and institutional frameworks. We used data from wave seven of the World Values Survey (WVS) and the Varieties of Democracy (V-Dem) dataset, covering diverse countries and time periods. We examined the interaction between emancipative values and a country's level of liberal democracy on life satisfaction using a random effects model accounting for country-level variations (random intercept and slope). The model was adjusted for individual-level variables, such as settlement size, income, and educational attainment, as well as country-level factors like lagged income per capita and regional affiliation. This approach allowed us to investigate the direct effect of emancipative values across different democratic contexts. Our final dataset included N = 76,702 participants across 58 countries. Countries with the highest levels of liberal democracy were also the most economically prosperous. Individuals in democratic nations were older, resided predominantly in larger settlements, demonstrated the highest levels of educational attainment, reported the highest incomes, and expressed the strongest adherence to emancipative values. Our random effect model revealed a negative main effect of emancipative values on life satisfaction. A significant positive interaction between democracy and emancipative values was also identified. Less democratic countries demonstrated a negative effect of emancipative values while no significant effect was observed in more democratic contexts. Overall, our study challenges the notion that emancipative values universally enhance life satisfaction, highlighting the significance of cultural and institutional congruence in assessing how personal values affect well-being. These findings emphasize the importance of considering contextual factors when examining the relationship between individual values and life satisfaction across diverse political and cultural landscapes.

**Data availability statement:** The data that support the findings of this study and the code used to analyse the data are available on a github repository https://github.com/gbarbalat/WVS-emancip-democracy-satisf

**Funding:** The author(s) received no specific funding for this work.

**Competing interests:** NO authors have competing interests.

## Introduction

Emancipative values advocate for the expansion of personal freedoms, self-expression, equality and autonomy. These values have been theorized to enhance life satisfaction, primarily through fostering a growing sense of empowerment and creating a wide array of opportunities for individuals [1]. Such empowerment manifests in various aspects of life, including the exercise of property rights, access to education, freedom of expression, increased tolerance and interpersonal trust, and active civic participation, all of which contribute to personal and societal development [2].

The human development model proposes that emancipative values foster a connection between agency and life satisfaction, but only after an expansion in the range of opportunities available to individuals [2]. This aligns with broader ideas that the fulfillment and impact of values on well-being are not universal but depend heavily on their congruence with societal norms [3–5]. Even at low levels, an alignment between individual and societal values can positively influence subjective well-being. Conversely, a mismatch between personal values and societal expectations may lead to cultural estrangement and diminished well-being. From an institutional approach, individual characteristics would contribute to subjective well-being to the extent that macro-level conditions are favorable for people with such characteristics. In essence, the relationship between emancipative values and well-being may be more nuanced than initially theorized, depending on the socio-cultural and institutional context in which these values operate.

Democracy, as a system of governance, embodies many of the principles inherent with emancipative values at a societal level. It provides a broader range of free choices, civil liberties, and opportunities for civic participation, including the incorporation of minorities [6]. Democratic systems typically promote life satisfaction by facilitating better access to essential services such as healthcare [7,8] and education, and by enabling personal freedoms as well as more equitable resource distribution compared to non-democratic practices. Similarly to emancipative values however, the relationship between democratic governance and subjective well-being may be nuanced and context-dependent. The beneficial impacts of democracy are frequently intertwined with a nation's economic prosperity. A country's wealth plays a crucial role in realizing the full potential of democratic governance, as substantial resources are essential for establishing and maintaining robust public institutions, infrastructure, and social services [9]. Likewise, the advantages of democracy and economic growth are not uniformly distributed across society. An individual's income level significantly influences their ability to access and benefit from public services and economic opportunities [10].

The interplay between individual-level emancipative values and country-level democracy in shaping well-being became evident during the democratization process in Central and Eastern Europe, where systemic changes led to significant societal disruptions and shifts in individual values [11]. This raises an important question about the empirical relationship between political beliefs and life satisfaction when expectations and reality are misaligned, which, to our knowledge, remains underexplored.

At this point, it is important to recognize that while emancipative values are closely linked to democracy, they emerge at different levels: emancipative values arise

at the individual level, whereas democracy operates at the level of governance or the state. In this sense, democracy provides the framework or enabling environment in which emancipative values can be expressed and realized. However, individuals living in democratic societies may not always hold emancipative values, and conversely, individuals in less democratic countries can still embrace and uphold emancipative values. Likewise, while there appears to be an obvious link between democracy and emancipative values, this connection is nuanced and influenced by several factors. To begin with, it is important to clarify what is meant by democracy, as it exists at different levels that may relate to emancipative values in varying degrees. As Coppedge et al. note, there is no consensus on a single definition of democracy beyond the basic idea of "rule by the people" [12]. In practice, most existing democracies are to some degree liberal and electoral, characterized by the rule of law, civil liberties, judicial independence, and free and fair elections of representatives. As conceptualized by the V-Dem project, there are three other key dimensions of democracy: participatory, deliberative, and egalitarian democracy, each capturing distinct but interconnected aspects of democratic life [12].

Recent research highlights that emancipative values drive a "critical-liberal" demand for democracy, where individuals with strong emancipative values insist on democratic freedoms and hold high standards for democratic quality, including making rulers truly responsive to citizens' rights and freedoms [13], Likewise, electoral democracy serves as the essential mechanism that institutionalizes this responsiveness. Emancipative values motivate people to insist on genuine electoral competition, political pluralism, and the rejection of authoritarian or fraudulent practices [13].

Other forms of democracy – deliberative, participatory, and egalitarian – also connect deeply with emancipative values by emphasizing broader citizen engagement, inclusive decision-making, and equality. For instance, John Dewey viewed democracy as a moral and social association in which individuals realize their full potential through active involvement in their communities and institutions [14]. Dewey believed that such participation and deliberation are fundamental to individual freedom, extending beyond government into all areas of social life, fostering collaborative problem-solving and critical thinking.

Among the five dimensions of democracy, liberal democracy stands out as the most relevant domain for exploring emancipative values. As mentioned above, emancipation (autonomy, equality, choice, and voice) emphasizes values that align closely with liberal democracy (protection of rights, rule of law, and freedom to pursue one's true self). Moreover, liberal democracy is perhaps the most tangible and visible form of democracy in our contemporary world, currently being seen as under threat, or alternatively as too hegemonic [15].

In the current study, we aimed to investigate whether the effect of emancipative values on life satisfaction is moderated by a country's level of democracy. Understanding the relationship between emancipative values, political systems, and individual well-being, beyond the simplistic assumptions about the universal benefits of emancipative values or democratic systems, is crucial for a nuanced comprehension of human development and societal progress. Such knowledge is particularly relevant as policymakers shape democratic institutions, and in times where democracy is highly questioned.

We predicted that the democratic context would play a key role in shaping the effect of emancipative values on life satisfaction, manifesting as an interaction between liberal democracy and these values. The rising expectations model of revolution offers valuable insights to this hypothesis [16]. In essence, the model suggests that citizens become frustrated when there is a gap between their heightened expectations for a better life and their actual experiences (with increasing gaps potentially sparking revolutionary unrest). Applied to our research, this implies that in countries with low levels of democracy, individuals with high emancipative values may experience lower life satisfaction due to these unmet expectations. In turn, increasing levels of democracy might not necessarily lead to higher life satisfaction among those with the highest emancipative values, as they may still feel discontented that their ideals remain unfulfilled.

## Methods

### Data

**Databases.** We relied on the 7th wave of the World Values Survey (WVS) time-series dataset covering the period 2017–2022. The WVS is a publicly available dataset that offers individual-level data on both attitudes towards democracy

and well-being [17]. The 7<sup>th</sup> wave of the WVS was conducted in 66 countries, covering about 90% of the world's population. Samples were nationally representative. The sample size ranges from 1004 to 4018 respondents in each country.

We merged the WVS dataset with the Varieties of Democracy (V-Dem) dataset. V-Dem is a comprehensive project that aims to conceptualize and measure democracy in a multidimensional way, recognizing that democracy can take various forms and is not limited to just electoral processes. V-Dem typically gathers data from five experts per country-year observation, drawing from a pool of over 4,000 country experts. These experts provide judgments on various concepts. V-Dem then employs a series of model and data transformation processes to provide valid and reliable estimates of concepts related to democracy, while accounting for uncertainty and potential biases in expert judgments. The V-Dem dataset is also a publicly available dataset, which we obtained from the *vdemdata* R package.

**Outcome variable: life satisfaction.** Our primary outcome variable was the self-rated measure of life satisfaction obtained from the WVS. This indicator was initially measured on a 10-point Likert scale, which we transformed into a numeric variable.

**Individual-level emancipative values and country-level index of liberal democracy.** We used the concept of emancipative values as measured by the WVS [18]. Emancipative values are a set of cultural values that prioritize universal human freedoms and individual choices. They are characterized by four key components:

- Voice: emphasizing freedom of speech and citizens' say in society;

- Autonomy: valuing control over one's own life;

- Choice: prioritizing freedom in personal life decisions (e.g., abortion and sexual orientation);

- Equality: stressing equal opportunities and non-discrimination.

Emancipative values are measured using 12 items from the WVS, with scores typically ranging from 0 to 1. Higher scores indicate stronger emphasis on emancipative values.

Country-level democracy, obtained from the V-dem database is measured as a multiple time-point variable on a scale from 0 to 1 (respectively a low vs. a high level of democracy). We focused primarily on liberal aspects of democracy, which emphasize the protection of individual and minority rights against potential tyranny of the majority and state repression through constitutionally-protected civil liberties, strong rule of law, and effective checks and balances. These elements are closely aligned with emancipative values.

**Covariates.** Both individual and country-level factors were included in our analysis as influencing life satisfaction. Individual aspects included: age; gender; educational attainment; settlement size; marital status; being a citizen of the country; living with parents in the household; number of children living at home; educational level; employment status; perception of income level; religious denomination; frequency of not having had enough food to eat over the past 12 months; frequency of having felt unsafe from crime in own home over the past 12 months; frequency of not having needed medicine or treatment over the past 12 months; number of active memberships (as a proxy for social activities); year of measurement; and month of measurement.

In terms of country-level factors, we included geographical region as a key variable that often encapsulates many of the cultural, historical and institutional factors thought to impact both country-level democracy and individual life satisfaction, e.g., shared historical experiences and cultural values among neighboring countries, regional diffusion of political systems and ideas, common economic and social challenges. We also included Lag-Distributed Income (LDI) per capita, a component of the Socio-demographic Index (SDI) used in the Global Burden of Disease (GBD) studies [19]. It represents a smoothed version of Gross Domestic Product (GDP) per capita that accounts for short-term fluctuations. The LDI is calculated using a weighted average of GDP per capita from previous years, with more recent years given higher weight, providing a more stable measure of long-term economic development. The LDI is adjusted for inflation and differences in cost of living between countries (using purchasing power parity).

Note that some of these covariates may be influenced by both individuals' emancipative values and countries' level of liberal democracy (e.g., income, settlement size, educational attainment, etc...). We included these variables in our final model to measure the association between emancipative values and life satisfaction across various levels of democracy, while controlling for these dimensions. This approach allowed us to isolate the effect of emancipative values and liberal democracy on life satisfaction beyond the influence of these other factors.

**Missing data.** Of the 97,220 participants included in the survey, we selected those living in countries with recorded indices of liberal democracy according to V-Dem. We further restricted our sample to participants aged 18–65 years. We then removed observations with more than 30% missing values. For the remaining 76,702 participants from 58 countries, we imputed missing data using the predictive mean matching algorithm from the R *mice* package.

## Ethics statement

This study utilized publicly available data from the WVS. As the data were collected by the WVS organization following their established ethical guidelines and consent procedures, additional informed consent was not required for this secondary analysis.

Participants in the original WVS provided consent according to the survey's protocols. For this study, which involves secondary analysis of publicly available data, individual participant consent was not obtained. The use of this data complies with the terms and conditions set forth by the WVS for research purposes.

## Analysis

To examine the varying effects of emancipative values across different levels of liberal democracy, we conducted a linear mixed effects regression using the following equation:

$$Y_{ij} = \alpha + \sum_k \beta_k Z_{ik} + \sum_l \gamma_l X_{jl} + \delta Dem_j * Emancip_{ij} + \zeta YearPrior_j + u_{0j} + u_{1j}Emancip_{ij} + \in_{ij}$$

$$u_{0j} \sim N\left(0, \sigma_{u_0}^2\right)$$
$$u_{1j} \sim N\left(0, \sigma_{u_1}^2\right)$$
$$\in_{ij} \sim N\left(0, \sigma_{\in}^2\right),$$

where $Y_{ij}$ is the life satisfaction for individual $i$ in country $j$; $\alpha$ is the overall intercept; $\sum_k \beta_k Z_{ik}$ represents the sum of individual-level predictors (including emancipative values), and their coefficients; $\sum_l \gamma_l X_{jl}$ represents the sum of country-level predictors (including liberal democracy), and their coefficients; $\delta Dem_j * Emancip_{ij}$ represents the interaction between the level of liberal democracy of country $j$ and emancipative values of individual $i$, and its coefficient; $\zeta YearPrior_j$ denotes the number of years for which liberal democracy and LDI measurements were used prior to assessing life satisfaction, and its coefficient (applicable only when *YearPrior* > 1); $u_{0j}$ is the random intercept for each country, which follows a normal distribution with mean 0 and variance $\sigma_{u0}^2$; $u_{1j}Emancip_{ij}$ is the random slope for the emancipation factor, which follows a normal distribution with mean 0 and variance $\sigma_{u1}^2$; $\in_{ij}$ is the residual error term, which follows a normal distribution with mean 0 and variance $\sigma_{\in}^2$.

Note that including country as a random factor allowed both the intercept and the effect of emancipative values on life satisfaction to vary by country. This approach accounted for the likelihood that observations within the same country are more similar to each other than to observations in other countries, thus addressing potential clustering effects in the data.

Also note that we conducted our main analysis using *YearPrior* = 1, and performed additional analyses for *YearPrior* values ranging from 2 to 10 years to assess the robustness of our findings across different time lags.

## Results

Individual and country characteristics, categorized across four ascending levels of liberal democracy, are reported in **Table 1**. Countries with the highest levels of liberal democracy were also the most economically prosperous. These democratic nations

**Table 1. Socio-demographic characteristics of the participants.**

| Variable | Low level of liberal democracy N = 26,722[1] | Mid-low level of liberal democracy N = 19,417[2] | Mid-high level of liberal democracy N = 12,412[3] | High level of liberal democracy N = 18,151[4] | Total N = 76,702 |
|---|---|---|---|---|---|
| Lib. Dem. Index | 0.12 (0.06) | 0.35 (0.06) | 0.57 (0.06) | 0.79 (0.05) | 0.41 (0.27) |
| LDI per capita | 12,145 (6,847) | 13,798 (20,030) | 14,245 (6,643) | 36,852 (11,362) | 18,750 (16,056) |
| Emancipative values | 0.35 (0.14) | 0.38 (0.15) | 0.40 (0.15) | 0.59 (0.18) | 0.42 (0.18) |
| Life satisfaction | 6.77 (2.39) | 7.20 (2.37) | 7.16 (2.32) | 7.15 (1.88) | 7.03 (2.27) |
| Sex | | | | | |
| Male | 12,892 (48%) | 9,422 (49%) | 5,570 (45%) | 8,285 (46%) | 36,169 (47%) |
| Female | 13,830 (52%) | 9,995 (51%) | 6,842 (55%) | 9,866 (54%) | 40,533 (53%) |
| Marital Status | | | | | |
| Married | 16,990 (64%) | 10,886 (56%) | 6,840 (55%) | 8,580 (47%) | 43,296 (56%) |
| Living together as married | 1,086 (4.1%) | 1,558 (8.0%) | 1,377 (11%) | 2,702 (15%) | 6,723 (8.8%) |
| Divorced, separated or widowed | 2,154 (8.1%) | 1,512 (7.8%) | 1,208 (9.7%) | 2,012 (11%) | 6,886 (9.0%) |
| Single/never married | 6,492 (24%) | 5,461 (28%) | 2,987 (24%) | 4,857 (27%) | 19,797 (26%) |
| Citizenship | | | | | |
| No | 96 (0.4%) | 228 (1.2%) | 91 (0.7%) | 840 (4.6%) | 1,255 (1.6%) |
| Yes | 26,626 (99.6%) | 19,189 (98.8%) | 12,321 (99.3%) | 17,311 (95.4%) | 75,447 (98.4%) |
| Living with parents | | | | | |
| No | 16,482 (62%) | 11,481 (59%) | 8,455 (68%) | 15,177 (84%) | 51,595 (67%) |
| Yes | 10,240 (38%) | 7,936 (41%) | 3,957 (32%) | 2,974 (16%) | 25,107 (33%) |
| Employment status | | | | | |
| Full time | 9,484 (35%) | 6,089 (31%) | 4,637 (37%) | 9,505 (52%) | 29,715 (39%) |
| Part time | 2,154 (8.1%) | 2,114 (11%) | 918 (7.4%) | 2,139 (12%) | 7,325 (9.5%) |
| Self employed | 4,492 (17%) | 4,043 (21%) | 2,428 (20%) | 1,532 (8.4%) | 12,495 (16%) |
| Retired | 1,243 (4.7%) | 566 (2.9%) | 545 (4.4%) | 1,000 (5.5%) | 3,354 (4.4%) |
| Housewife | 4,781 (18%) | 3,133 (16%) | 2,082 (17%) | 1,404 (7.7%) | 11,400 (15%) |
| Student | 1,834 (6.9%) | 1,396 (7.2%) | 739 (6.0%) | 1,055 (5.8%) | 5,024 (6.6%) |
| Unemployed | 2,476 (9.3%) | 1,843 (9.5%) | 978 (7.9%) | 1,143 (6.3%) | 6,440 (8.4%) |
| Other | 258 (1.0%) | 233 (1.2%) | 85 (0.7%) | 373 (2.1%) | 949 (1.2%) |
| Religious denomination | | | | | |
| Nil | 4,425 (17%) | 1,874 (9.7%) | 2,759 (22%) | 8,434 (46%) | 17,492 (23%) |
| Roman Catholic | 3,020 (11%) | 5,240 (27%) | 2,049 (17%) | 4,146 (23%) | 14,455 (19%) |
| Protestant | 702 (2.6%) | 2,503 (13%) | 1,076 (8.7%) | 1,738 (9.6%) | 6,019 (7.8%) |
| Orthodox/Jew | 1,717 (6.4%) | 902 (4.6%) | 1,721 (14%) | 1,456 (8.0%) | 5,796 (7.6%) |
| Muslim | 14,050 (53%) | 5,779 (30%) | 3,615 (29%) | 849 (4.7%) | 24,293 (32%) |
| Hindu | 171 (0.6%) | 1,205 (6.2%) | 103 (0.8%) | 143 (0.8%) | 1,622 (2.1%) |
| Buddhist | 2,016 (7.5%) | 1,426 (7.3%) | 721 (5.8%) | 320 (1.8%) | 4,483 (5.8%) |
| Other Christian | 581 (2.2%) | 221 (1.1%) | 158 (1.3%) | 698 (3.8%) | 1,658 (2.2%) |
| Other | 40 (0.1%) | 267 (1.4%) | 210 (1.7%) | 367 (2.0%) | 884 (1.2%) |
| Age | | | | | |
| 18-24 | 4,684 (18%) | 3,938 (20%) | 2,029 (16%) | 2,032 (11%) | 12,683 (17%) |
| 25−24 | 6,952 (26%) | 5,415 (28%) | 3,018 (24%) | 3,937 (22%) | 19,322 (25%) |
| 35-44 | 6,222 (23%) | 4,356 (22%) | 2,848 (23%) | 4,051 (22%) | 17,477 (23%) |
| 45-54 | 5,118 (19%) | 3,246 (17%) | 2,501 (20%) | 4,083 (22%) | 14,948 (19%) |
| 55-64 | 3,746 (14%) | 2,462 (13%) | 2,016 (16%) | 4,048 (22%) | 12,272 (16%) |

*(Continued)*

**Table 1.** (Continued)

| Variable | Low level of liberal democracy N = 26,722[1] | Mid-low level of liberal democracy N = 19,417[2] | Mid-high level of liberal democracy N = 12,412[3] | High level of liberal democracy N = 18,151[4] | Total N = 76,702 |
|---|---|---|---|---|---|
| Settlement size | | | | | |
| Under 2,000 | 2,517 (9.4%) | 3,178 (16%) | 1,050 (8.5%) | 1,472 (8.1%) | 8,217 (11%) |
| 2-5,000 | 2,454 (9.2%) | 2,970 (15%) | 1,700 (14%) | 941 (5.2%) | 8,065 (11%) |
| 5-10,000 | 2,257 (8.4%) | 2,006 (10%) | 1,344 (11%) | 855 (4.7%) | 6,462 (8.4%) |
| 10-20,000 | 2,099 (7.9%) | 1,346 (6.9%) | 920 (7.4%) | 1,312 (7.2%) | 5,677 (7.4%) |
| 20-50,000 | 2,942 (11%) | 1,776 (9.1%) | 1,073 (8.6%) | 2,631 (14%) | 8,422 (11%) |
| 50-100,000 | 3,120 (12%) | 1,112 (5.7%) | 904 (7.3%) | 2,198 (12%) | 7,334 (9.6%) |
| 100-500,000 | 5,620 (21%) | 2,194 (11%) | 1,528 (12%) | 4,427 (24%) | 13,769 (18%) |
| 500,000+ | 5,713 (21%) | 4,835 (25%) | 3,893 (31%) | 4,315 (24%) | 18,756 (24%) |
| Nb of children | | | | | |
| 0 | 7,778 (29%) | 6,044 (31%) | 3,283 (26%) | 6,919 (38%) | 24,024 (31%) |
| 1 | 4,964 (19%) | 3,084 (16%) | 2,373 (19%) | 3,242 (18%) | 13,663 (18%) |
| 2 | 6,596 (25%) | 4,232 (22%) | 3,637 (29%) | 4,879 (27%) | 19,344 (25%) |
| 3 | 3,611 (14%) | 2,838 (15%) | 1,823 (15%) | 1,981 (11%) | 10,253 (13%) |
| 4 | 1,891 (7.1%) | 1,619 (8.3%) | 748 (6.0%) | 717 (4.0%) | 4,975 (6.5%) |
| 5+ | 1,882 (7.0%) | 1,600 (8.2%) | 548 (4.4%) | 413 (2.3%) | 4,443 (5.8%) |
| Education level | | | | | |
| Lower | 10,790 (40%) | 6,570 (34%) | 4,038 (33%) | 2,471 (14%) | 23,869 (31%) |
| Middle | 8,047 (30%) | 6,818 (35%) | 5,100 (41%) | 7,295 (40%) | 27,260 (36%) |
| Upper | 7,885 (30%) | 6,029 (31%) | 3,274 (26%) | 8,385 (46%) | 25,573 (33%) |
| Scale of incomes | | | | | |
| Lower step | 2,074 (7.8%) | 1,589 (8.2%) | 1,416 (11%) | 759 (4.2%) | 5,838 (7.6%) |
| Second step | 1,485 (5.6%) | 1,191 (6.1%) | 783 (6.3%) | 832 (4.6%) | 4,291 (5.6%) |
| Third step | 3,166 (12%) | 2,015 (10%) | 1,274 (10%) | 1,761 (9.7%) | 8,216 (11%) |
| Fourth step | 4,010 (15%) | 2,487 (13%) | 1,552 (13%) | 2,493 (14%) | 10,542 (14%) |
| Fifth step | 6,864 (26%) | 4,855 (25%) | 3,076 (25%) | 3,734 (21%) | 18,529 (24%) |
| Sixth step | 4,132 (15%) | 2,791 (14%) | 1,832 (15%) | 3,155 (17%) | 11,910 (16%) |
| Seventh step | 2,736 (10%) | 2,222 (11%) | 1,392 (11%) | 2,783 (15%) | 9,133 (12%) |
| Eigth step | 1,354 (5.1%) | 1,382 (7.1%) | 685 (5.5%) | 1,467 (8.1%) | 4,888 (6.4%) |
| Nineth step | 409 (1.5%) | 375 (1.9%) | 156 (1.3%) | 624 (3.4%) | 1,564 (2.0%) |
| Highest step | 492 (1.8%) | 510 (2.6%) | 246 (2.0%) | 543 (3.0%) | 1,791 (2.3%) |
| Gone without enough food to eat (last 12 months) | | | | | |
| Often | 1,401 (5.2%) | 918 (4.7%) | 706 (5.7%) | 432 (2.4%) | 3,457 (4.5%) |
| Sometimes | 3,458 (13%) | 3,019 (16%) | 1,824 (15%) | 1,367 (7.5%) | 9,668 (13%) |
| Rarely | 4,922 (18%) | 3,822 (20%) | 1,887 (15%) | 2,235 (12%) | 12,866 (17%) |
| Never | 16,941 (63%) | 11,658 (60%) | 7,995 (64%) | 14,117 (78%) | 50,711 (66%) |
| Felt unsafe from crime in your own home (last 12 months) | | | | | |
| Often | 1,112 (4.2%) | 1,120 (5.8%) | 732 (5.9%) | 775 (4.3%) | 3,739 (4.9%) |
| Sometimes | 3,188 (12%) | 2,854 (15%) | 1,618 (13%) | 2,202 (12%) | 9,862 (13%) |
| Rarely | 4,933 (18%) | 3,712 (19%) | 1,689 (14%) | 3,460 (19%) | 13,794 (18%) |
| Never | 17,489 (65%) | 11,731 (60%) | 8,373 (67%) | 11,714 (65%) | 49,307 (64%) |
| Gone without needed medicine or treatment that you needed (last 12 months) | | | | | |
| Often | 1,881 (7.0%) | 1,215 (6.3%) | 942 (7.6%) | 591 (3.3%) | 4,629 (6.0%) |
| Sometimes | 4,629 (17%) | 3,639 (19%) | 2,399 (19%) | 1,985 (11%) | 12,652 (16%) |
| Rarely | 5,616 (21%) | 4,039 (21%) | 2,224 (18%) | 2,473 (14%) | 14,352 (19%) |
| Never | 14,596 (55%) | 10,524 (54%) | 6,847 (55%) | 13,102 (72%) | 45,069 (59%) |
| Life satisfaction | 7.00 (5.00, 9.00) | 7.00 (6.00, 9.00) | 7.00 (6.00, 9.00) | 7.00 (6.00, 8.00) | 7.00 (6.00, 9.00) |

*(Continued)*

**Table 1.** (Continued)

| Variable | Low level of liberal democracy N = 26,722[1] | Mid-low level of liberal democracy N = 19,417[2] | Mid-high level of liberal democracy N = 12,412[3] | High level of liberal democracy N = 18,151[4] | Total N = 76,702 |
|---|---|---|---|---|---|
| Active memberships | | | | | |
| 0 | 16,964 (63%) | 10,644 (55%) | 6,994 (56%) | 9,563 (53%) | 44,165 (58%) |
| 1 | 5,294 (20%) | 4,147 (21%) | 2,683 (22%) | 4,420 (24%) | 16,544 (22%) |
| 2 | 1,910 (7.1%) | 2,036 (10%) | 1,200 (9.7%) | 2,157 (12%) | 7,303 (9.5%) |
| 3+ | 2,554 (9.6%) | 2,590 (13%) | 1,535 (12%) | 2,011 (11%) | 8,690 (11%) |
| Region | | | | | |
| Eastern Europe | 5,023 (19%) | 2,148 (11%) | 3,452 (28%) | 950 (5.2%) | 11,573 (15%) |
| Latin America | 2,209 (8.3%) | 5,689 (29%) | 3,705 (30%) | 2,897 (16%) | 14,500 (19%) |
| North Africa and the Middle East | 9,481 (35%) | 1,098 (5.7%) | 1,074 (8.7%) | 0 (0%) | 11,653 (15%) |
| Sub–Saharan Africa | 2,303 (8.6%) | 2,450 (13%) | 0 (0%) | 0 (0%) | 4,753 (6.2%) |
| Western countries[5] | 0 (0%) | 0 (0%) | 0 (0%) | 13,427 (74%) | 13,427 (18%) |
| Eastern Asia | 2,711 (10%) | 0 (0%) | 1,138 (9.2%) | 877 (4.8%) | 4,726 (6.2%) |
| South–Eastern Asia | 3,829 (14%) | 3,843 (20%) | 3,043 (25%) | 0 (0%) | 10,715 (14%) |
| Southern Asia | 1,166 (4.4%) | 4,189 (22%) | 0 (0%) | 0 (0%) | 5,355 (7.0%) |
| Year | | | | | |
| 2017 | 2,909 (11%) | 3,112 (16%) | 2,959 (24%) | 4,270 (24%) | 13,250 (17%) |
| 2018 | 12,101 (45%) | 5,656 (29%) | 5,893 (47%) | 3,371 (19%) | 27,021 (35%) |
| 2019 | 2,267 (8.5%) | 4,998 (26%) | 2,612 (21%) | 2,375 (13%) | 12,252 (16%) |
| 2020 | 4,830 (18%) | 2,163 (11%) | 0 (0%) | 3,257 (18%) | 10,250 (13%) |
| 2021 | 3,406 (13%) | 2,215 (11%) | 948 (7.6%) | 0 (0%) | 6,569 (8.6%) |
| 2022/2023 | 1,209 (4.5%) | 1,273 (6.6%) | 0 (0%) | 4,878 (27%) | 7,360 (9.6%) |
| Month | | | | | |
| January | 2,329 (8.7%) | 6,622 (34%) | 0 (0%) | 3,046 (17%) | 11,997 (16%) |
| February | 2,303 (8.6%) | 0 (0%) | 0 (0%) | 950 (5.2%) | 3,253 (4.2%) |
| March | 3,787 (14%) | 0 (0%) | 0 (0%) | 0 (0%) | 3,787 (4.9%) |
| April | 1,275 (4.8%) | 0 (0%) | 1,074 (8.7%) | 5,208 (29%) | 7,557 (9.9%) |
| May | 1,108 (4.1%) | 1,243 (6.4%) | 2,393 (19%) | 879 (4.8%) | 5,623 (7.3%) |
| June | 3,396 (13%) | 2,371 (12%) | 3,043 (25%) | 0 (0%) | 8,810 (11%) |
| July-August | 2,711 (10%) | 1,032 (5.3%) | 855 (6.9%) | 1,910 (11%) | 6,508 (8.5%) |
| September | 0 (0%) | 1,170 (6.0%) | 1,538 (12%) | 1,770 (9.8%) | 4,478 (5.8%) |
| October | 1,173 (4.4%) | 0 (0%) | 0 (0%) | 4,388 (24%) | 5,561 (7.3%) |
| November | 3,770 (14%) | 3,593 (19%) | 2,371 (19%) | 0 (0%) | 9,734 (13%) |
| December | 4,870 (18%) | 3,386 (17%) | 1,138 (9.2%) | 0 (0%) | 9,394 (12%) |

[1]Bangladesh, China, Egypt, Ethiopia, Iran, Iraq, Jordan, Kazakhstan, Libya, Morocco, Malaysia, Nicaragua, Russia, Thailand, Tajikistan, Turkey, Uzbekistan, Venezuela, Vietnam, Zimbabwe

[2]Bolivia, Ecuador, Guatemala, India, Kenya, Kyrgyzstan, Lebanon, Maldives, Mexico, Myanmar, Nigeria, Pakistan, Philippines, Singapore, Ukraine

[3]Argentina, Armenia, Brazil, Colombia, Indonesia, South Korea, Mongolia, Romania, Tunisia

[4]Australia, Canada, Chile, Cyprus, Czech Republic, Germany, United Kingdom, Greece, Japan, Netherlands, New Zealand, Peru, Uruguay, United States

[5]Including Western Europe, North America, Australia and New Zealand

Lib. Dem.: Liberal democracy

LDI: Lag-distributed income per capita

had the oldest participants. Individuals in these countries predominantly resided in larger settlements; were less likely to be married; had fewer children on average; demonstrated the highest levels of educational attainment; reported the highest incomes; and showed the greatest engagement in activities. Notably, residents of these highly democratic countries also expressed the strongest adherence to emancipative values.

Our random effect model retrieved an overall negative effect of emancipative values on life satisfaction (b = −0.88; p < 0.001) but no significant effect of liberal democracy (b = −0.37; p = 0.41). However, there was a significant positive interaction between democracy and emancipative values (b = 1.04; p = 0.024). Standardized mean difference (Cohen's d) for the interaction between liberal democracy and emancipative values was of 0.46. As a point of comparison, other variables were predictive of life satisfaction such as income level (b = 0.17; p < 0.001; d = 0.07), having no food (b = 0.29; p < 0.001; d = 0.13), being involved in group memberships (b = 0.10; p < 0.001; d = 0.04), settlement size (b = 0.09; p < 0.001; d = 0.04), or being divorced, separated or widowed (b = −0.34; p < 0.001; d = −0.15).

To further examine the positive interaction between democracy and emancipative values, we conducted a post-hoc Johnson-Neyman analysis. This analysis identified the specific levels of democracy at which emancipative values significantly predicted life satisfaction. There was a continuous increase in the mean estimate of emancipative values' effect on life satisfaction across societies ranging from least to most democratic. Moreover, we found a significantly negative effect of emancipative values on life satisfaction at low levels of democracy (liberal democracy index lower than 0.50), but no significant effect at higher levels (**Fig 1**).

**S1 Fig** displays the interaction between emancipative values and liberal democracy, taking into account democracy levels from two to ten years prior to the survey year. The positive interaction between emancipative values and liberal democracy remained significant when considering data from up to four years prior to the survey year. The pattern of significance across different levels of democracy was consistent, showing a negative effect of emancipative values in less democratic contexts and no effect in more democratic contexts.

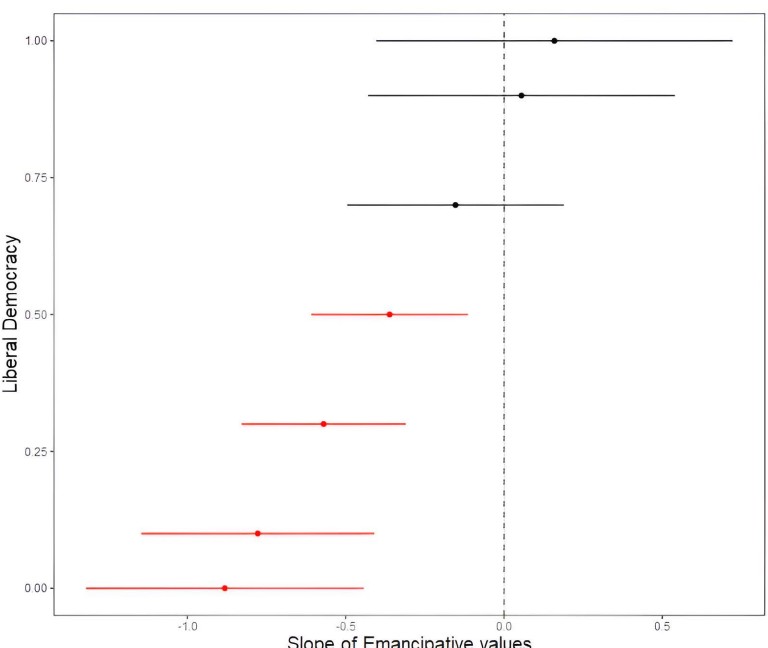

**Fig 1. Effect of emancipative values on life satisfaction at different levels of liberal democracy.** Effect of emancipative values on life satisfaction across varying levels of liberal democracy, based on a model using democracy measures from one year prior to the survey year.

## Discussion

In this study, we investigated how the relationship between emancipative values and life satisfaction is moderated by the democratic context in which individuals live. While emancipative values are often associated with increased well-being, our findings reveal a more nuanced picture. After adjusting for various individual- and country-level factors, we observed a negative direct effect of emancipative values on life satisfaction. This negative effect appears to be moderated by the democratic context, as we found no significant effect in societies with higher levels of democracy. When interpreting our findings, it is important to remember that our statistical estimates were derived from a model that controlled for other key predictors of life satisfaction (i.e., "all other things being equal"). In particular, we must emphasize that ensuring basic survival remains a fundamental concern, and that life satisfaction depends on far more than just the interaction between emancipative values and democracy. Nonetheless, the effect size for the interaction between emancipative values and the level of democracy approached a moderate magnitude and was, in absolute terms, larger than other statistically significant predictors of life satisfaction, such as income level, food security, and involvement in active memberships.

Our results reveal a complex relationship between the level of democracy, emancipative values, and life satisfaction, challenging conventional wisdom regarding the universally positive impact of emancipative values. This complexity aligns with recent findings in the field. For instance, Krys et al. demonstrated that while there is a positive zero-order correlation between self-expression and happiness or subjective well-being, deeper analysis suggests that this association may be primarily driven by "open society attitudes" that foster care, trust, and interdependence [20]. When controlling for these open-society attitudes, more individualistic aspects of emancipation – such as opportunity, independence, and right to privacy – emerged as negative predictors of life satisfaction. Similarly, Yu's work on secularization, a set of values emphasizing individual freedoms and autonomy in the context of separating religion from public affairs, found a negative relationship with life satisfaction [21]. Yu concluded that the positive relationship found in previous studies might be oversimplified, further underscoring the nuanced nature of these interactions and the need for more sophisticated models to understand the true impact of secular values on well-being.

Our findings offer a nuanced perspective on Welzel's theory of human development, which posits that emancipative values lead to higher life satisfaction once basic survival needs are met [2]. After controlling for survival-related variables (such as lagged distributed income per capita, individual income, and food security), we observed a positive interaction between emancipative values and liberal democracy. This interaction suggests that the theory's predictions hold relatively true in democratic societies when using less democratic societies as a reference point. However, our results also indicate that the relationship between emancipative values and life satisfaction is more complex than initially theorized, varying with the level of democracy and potentially influenced by other societal factors.

Our results highlights the importance of considering the democratic context when examining the impact of emancipative values on well-being. In less democratic environments, emancipative values may highlight a disconnect between individual aspirations and societal realities, potentially leading to frustration and social unrest as individuals struggle to align their beliefs with prevailing norms. Therefore, our results fit well with the rising expectations model of revolution which we briefly presented in the Introduction. Related to the model is the concept of relative deprivation, i.e., the discrepancy in what people expect to achieve and what they actually achieve [22]. According to this concept, it is this discrepancy, rather than absolute deprivation itself, that gives rise to dissatisfaction. The absence of a negative effect in more democratic societies suggests that as societies become more democratic, they may better accommodate citizens with emancipative values, aligning institutional structures with evolving societal values to enhance overall well-being.

Interestingly, we did not find a positive effect of emancipative values in highly democratic societies, possibly because individuals with less emancipative values can still thrive in such environments. Indeed, the emphasis on personal freedom and respect for others' boundaries in highly democratic countries likely allows those who differ from the majority

to feel comfortable. The non-significant effect of emancipation in highly democratic societies may also be attributed to the heightened tolerance for ambiguity and uncertainty associated with emancipative mindsets, which could inherently generate stress and potentially offset some of the benefits [23–25]. Another hypothesis to explain this negative finding is that increasing freedoms raise expectations, and greater awareness of remaining restrictions or repression can feel intolerable, and in turn would not significantly alleviate dissatisfaction. This dynamic may occur because of what has been conceptualized as the "Tocqueville effect" [26]. When freedom and emancipation become dominant ideals in democratic societies, any visible repression clashes sharply with those ideals, fueling discontent. In addition, Tocqueville warns that although formal liberties may exist, the growth of centralized power and bureaucratic control can subtly undermine genuine freedom, leading to a different kind of dissatisfaction, here not merely because rising expectations are unmet, but because the exercise of freedom is constrained even as its appearance remains [27].

Finally, this intricate relationship between democracy, emancipative values, and life satisfaction appears more pronounced when considering recent years of democracy rather than long-term trends. This suggests that the impact of emancipative values on life satisfaction is more responsive to recent changes in democratic quality rather than long-term democratic traditions. Looking ahead, what might happen as nations become even more democratic? One possibility is that the relationship between emancipative values and life satisfaction could eventually turn significantly positive, particularly as democratic innovations – such as citizen deliberation, direct democracy, or participatory governance – have demonstrated substantial societal benefits [28,29]. Alternatively, following the "Tocqueville effect" (as previously discussed), individuals with heightened emancipative values might continually seek further progress, meaning the relationship between these values and life satisfaction could remain non-significant even in highly democratic societies. Another possibility is that if democracy reaches exceptionally high levels, individuals with lower emancipative values may experience strong dissatisfaction. Further research is needed to determine whether this dynamic will manifest in the future.

## Limitations

Our study was not exempt of limitations. First, the WVS dataset is a time-series but it is not a panel dataset. No causal inference can be made from our results.

Second, our country random effects did not capture smooth spatial trends for emancipative values, democracy and life satisfaction. Our country random effect however supported differential meanings of emancipative values on life satisfaction across different societies [30].

Third, even though emancipative values are seen as a coherent measure combining tolerance, gender equality, political participation, and personal autonomy [31], the index has been criticized as mixing different value dimensions [30]. Taking this indicator as a whole might miss the nuance and complex interplay between different individual indicators, giving little indication on policy recommendations or theoretical conclusions on what is critically related to life satisfaction.

Fourth, others have argued for reclassifying emancipative values as policy preferences rather than values [31]. This reclassification is significant because preferences are often formulated at the time of choice and are more susceptible to contextual influences, while values are generally more stable and enduring [32]. Consequently, the effects found in the current study may not be temporally stable, as they could be influenced by changing contexts and circumstances rather than reflecting deeply held values.

Fifth, our random effects model assumed linearity and may not have captured potential non-linear dynamics or interactions. For example, the relationship between emancipative values and life satisfaction might follow a non-linear pattern (e.g., a bell curve) rather than a straightforward linear relationship. Interactions between emancipative values and factors such as age, gender, or income level could further nuance these relationships but were not explored here. Consequently, our findings should be interpreted as preliminary, highlighting the need for future studies to incorporate non-linear modeling and interaction effects.

## Conclusion

Our findings highlight the importance of considering the democratic institutional context in understanding the impact of emancipative values on life satisfaction. When considering the role of democratic governance, the relationship between emancipative values and life satisfaction appears to be more complex than previously thought, challenging the view that emancipative values are related to improved well-being. After adjusting for various individual- and country-level factors, we observed a negative direct effect of emancipative values on life satisfaction in societies with low levels of democracy but no effect in societies that have higher levels of democracy. These results underscore the necessity of a nuanced understanding of how emancipative values interact with democratic institutions, indicating that promoting these values may not universally lead to improved life satisfaction, particularly in less democratic contexts.

Several potential mechanisms may underlie our findings. For instance, a diminished sense of belonging could result from a mismatch between a country's level of democracy and an individual's emancipative values. Other important mechanisms to consider include feelings of unfairness, injustice, and national pride. To better understand these competing explanations, we recommend collecting longitudinal survey data that track emancipative values and potential mediators over time, alongside life satisfaction as the outcome. This would enable the application of causal mediation analysis to more rigorously test these pathways.

## Supporting information

**S1 Fig. Effect of emancipative values on life satisfaction at different levels of liberal democracy and historical contexts (2–10 years prior to survey).**
(DOCX)

## Author contributions

**Conceptualization:** Guillaume Barbalat, Nicolas Franck.

**Data curation:** Guillaume Barbalat.

**Formal analysis:** Guillaume Barbalat.

**Investigation:** Guillaume Barbalat.

**Methodology:** Guillaume Barbalat.

**Project administration:** Guillaume Barbalat, Nicolas Franck.

**Software:** Guillaume Barbalat.

**Supervision:** Nicolas Franck.

**Validation:** Nicolas Franck.

**Visualization:** Nicolas Franck.

**Writing – original draft:** Guillaume Barbalat.

**Writing – review & editing:** Guillaume Barbalat, Nicolas Franck.

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
