## [Decision Letter · Decision Letter 0]

20 Apr 2025

PONE-D-25-06086Effect of emancipative values on life satisfaction across different levels of democracy. A cross-national analysis of the World Values Survey.PLOS ONE

Dear Dr. Barbalat,

Thank you for submitting your manuscript to PLOS ONE. After careful consideration, we feel that it has merit but does not fully meet PLOS ONE’s publication criteria as it currently stands. Therefore, we invite you to submit a revised version of the manuscript that addresses the points raised during the review process.

We look forward to receiving your revised manuscript.

Kind regards,

Marco Improta

Academic Editor

PLOS ONE

3. In the online submission form, you indicated that [Data will be available upon request to the corresponding author].

Reviewers' comments:

Reviewer's Responses to Questions

**Comments to the Author**

1. Is the manuscript technically sound, and do the data support the conclusions?

Reviewer #1: Yes

Reviewer #2: Yes

2. Has the statistical analysis been performed appropriately and rigorously? 

Reviewer #1: N/A

Reviewer #2: Yes

3. Have the authors made all data underlying the findings in their manuscript fully available?

Reviewer #1: No

Reviewer #2: Yes

4. Is the manuscript presented in an intelligible fashion and written in standard English?

Reviewer #1: Yes

Reviewer #2: Yes

5. Review Comments to the Author

Reviewer #1: The study investigates the relationship between emancipative values, operationalized as values that advocate for personal freedoms, equality, and autonomy, and life satisfaction across different levels of democracy. Using data from the seventh wave of the World Values Survey (WVS) and the Varieties of Democracy (V-Dem) dataset, the authors analyze how the impact of emancipative values on life satisfaction varies depending on the democratic context of a country. The study employs a random effects model to account for country-level variations and adjusts for individual-level factors such as income, education, and settlement size, as well as country-level factors like lagged income per capita and regional affiliation. The findings reveal a negative main effect of emancipative values on life satisfaction, but a significant positive interaction between democracy and emancipative values, suggesting that the effect of these values is context-dependent. In less democratic countries, emancipative values have a negative impact on life satisfaction, while in more democratic contexts, no significant effect is observed. The results challenge the conventional assumption that emancipative values universally enhance life satisfaction, highlighting the importance of cultural and institutional congruence, and suggesting that the alignment between individual values and societal norms plays a crucial role in determining well-being. The study also notes that the relationship between emancipative values and life satisfaction is more responsive to recent changes in democratic quality rather than long-term democratic traditions.

The paper is rather short, with a very limited theoretical discussion. They refer to very limited availability of connected research. While one can concur with this fact, still a more dexterous work on the theoretical framework is missing, particularly with how the notion of democracy is presented. First, the paper makes no substantive mention to any of the relevant Political Science literature that could serve to contextualize how democracy is presented, beyond the concepts implied in the data selection. But authors have no advantage in saving the reader from such a discussion, leaving unanswered the important issue of a potential tautological issue, as the notion of emancipative values may be embedded in the very notion of democracy that they are using, which could explain the results in less developed countries. A more thorough discussion of what democracy is in this context is necessary, at least to remove this issue.

Regarding methodology, the explanatory power of the selected variables/ data is rather limited, as it is the linear analysis chosen. The authors acknowledge several limitations, saving the reader from the error of assuming automatic inferences. However, they state those limitations regarding several aspects, none of which related to very limited nature of linear analysis. For example, understood as systems, non-linear approaches may be more useful and less prone to frequentist approaches. Authors need to either more explicitly acknowledge this limitations in connection with the linear model selected, or enrich the analysis with eg non-linear, Bayesian or qualitative analysis.

In general, this is a very brief but well written and sufficiently crafted paper, although at its current stage more appropriate as a preliminary analysis report, than as a fully fledged paper. Hower, as long as author/s extend the theoretical analysis as indicated, remarkably to more clearly discern the relation between democracy and the studied values; and as long as it more explicitly states its statsitical limitations or improve its explanatory power, it may be considered for publication.

Reviewer #2: This article answers a broad question about values, society, and life satisfaction using two strong and complementary datasets, one country-level expert-coded dataset about democratic institutions/norms and another individual-level dataset about individuals' values and attitudes across the globe. The authors analyze this dataset thoughtfully and carefully, such as using multilevel analysis to avoid committing an ecological fallacy (that mixes levels in less rigorous ways).

The strength of the research is that it tests a simple proposition from a prior theory about emancipative values enhancing life satisfaction, so long as basic needs have been met. They find contrary evidence, in two respects. In democracies, the values have no bearing on satisfaction. In less democratic countries, they have a negative association, perhaps owing to being frustrated by the undemocratic political (and social? economic?) system in which they live. Their main takeaway is to stress that one's context matters when considering what beliefs lead to a satisfying life. It's a straightforward finding, which makes abundant sense, and it provides an interesting anomaly for the aforementioned theory to consider.

The weaknesses of the article seem relatively minor to me, in that each can be addressed by the authors with little difficulty. First, I found the theoretical setup too sketchy, even for what is meant to be a brief article. Did the authors truly have no hypothesis? Page 4 provides the closest account of the authors' purpose, which is simply to "investigate" this question. A lack of predictive directional hypothesis makes the main finding seem like the catch pulled out of a fishing expedition. If the authors had a clearer vision, they should state it here. If not, then they could add to the Limitations section that they had non-directional hypotheses and can present only post-hoc directional findings.

In either case, I'd like to see more theoretically grounded interpretation of these results. Surely they can find a stronger foundation or this pattern of outcomes. The "rising expectations" model of revolution came to mind, whereby people become frustrated by a govt as their expectations (for life, liberty, etc) rise--sparking revolution when the gap is too great. The authors don't say much about what these findings mean, nor how they fit into preexisting sociological work, political science research, and other scholarship.

A related suggestion is that the authors could provide better guidance for future research if they make clearer the theoretical stakes. I agree that scholars need to "explore the mechanisms" that could be at play in the results they find. But "exploration," per se, would repeat a weakness in here; that is, an exploratory approach suggests weak theoretical moorings for such future studies. Instead, I encourage the authors to posit potential rival mechanisms and how those might be tested. In a lab? In field experiments? In a longitudinal survey? If well conceived and described, this section of the article could be the key to inspiring well-designed future research that answers these questions more decisively.

A more speculative comment comes from looking at the data. I'd like the authors to speak more vividly about effect size. How do the effects they see compare to related statistical associations? Are these tiny, small, or larger coefficients? And so on. Looking at the figure, I also wonder if they could say more about their pattern of results. Currently, they write, "We found a significantly negative effect of emancipative values on 215 life satisfaction at low levels of democracy (liberal democracy index lower than 0.50), but no significant effect at 216 higher levels (Figure)." Fair enough, but researchers needn't only note what meets a p-value cutoff and what doesn't. (For that matter, some reviewers may be upset that the authors report as significant p values between .01 and .05, in spite of the enormous size of the dataset. That doesn't bother me, particularly for multilevel analysis, but there's dispute on this question.) What strikes me about the figure is that the mean effect size moves in a nearly linear fashion from least to most democratic societies. The way the authors describe it almost sounds like an s-shaped curve: suddenly associated below the .5 democracy cutoff, then just as suddenly non-sig above it. The pattern is more of a straight line. What does this mean? Could it be that as nations get *more* democratic, the positive association will, finally, become a significant one? The authors could note the democratic innovations literature and speculate out its payoff in the future.

6. PLOS authors have the option to publish the peer review history of their article (what does this mean? ). If published, this will include your full peer review and any attached files.

**Do you want your identity to be public for this peer review?** For information about this choice, including consent withdrawal, please see our Privacy Policy .

Reviewer #1: No

Reviewer #2: No

---

## [Author Response · Author response to Decision Letter 1]

7 May 2025

Please see Response to reviewers file

---

## [Editor Report · Decision Letter 1]

9 May 2025

Effect of emancipative values on life satisfaction across different levels of democracy. A cross-national analysis of the World Values Survey.

PONE-D-25-06086R1

Dear Dr. Barbalat,

We’re pleased to inform you that your manuscript has been judged scientifically suitable for publication and will be formally accepted for publication once it meets all outstanding technical requirements.

Kind regards,

Marco Improta

Academic Editor

PLOS ONE

---

## [Editor Report · Acceptance letter]

PONE-D-25-06086R1

PLOS ONE

Dear Dr. Barbalat,

I'm pleased to inform you that your manuscript has been deemed suitable for publication in PLOS ONE. Congratulations! Your manuscript is now being handed over to our production team.

Kind regards,

on behalf of

Dr. Marco Improta

Academic Editor

PLOS ONE